# Women Emotional, Cognitive and Physiological Modes of Coping with Daily Urban Environments: A Pilot Study

**DOI:** 10.3390/ijerph19138190

**Published:** 2022-07-04

**Authors:** Izhak Schnell, Basem Hijazi, Diana Saadi, Emanuel Tirosh

**Affiliations:** 1Geography and Human Environment Department, Exact Science Faculty, Tel Aviv University, Tel Aviv 6997801, Israel; basem.hijazi@gmail.com (B.H.); sdianaa@gmail.com (D.S.); 2The Bruce and Ruth Rappaport Faculty of Medicine Technion, Technical Institute of Technology, Haifa 3200003, Israel; emi.tirosh@gmail.com

**Keywords:** cognitive responses, emotional responses, physiological responses, environmental challenges, park, residential environment, city center environment

## Abstract

Studies on the effect of urban environments on human risk to health and well-being tend to focus on either physiological or cognitive and emotional effects. For each of these effects, several indicators have been proposed. They are determined either by a physiological-emotional theory or by a cognitive theory of direct attention. However, the interrelationships between these indices have not been thoroughly investigated in environmental contexts. Recently, a neuro-visceral model that incorporates all three aspects has been proposed. The present article focuses on understanding the mechanism of coping with urban environments. More specifically, we analyze the interrelations among nine of the more commonly used indices that represent the physiological, emotional and cognitive aspects of coping with urban environments. The data were collected in the following four environments: home, park, city center and residential area. The participants were 72 healthy, middle-class mothers with either high school or postgraduate education. They wherein their fertile age (20–35) with average Body Mass Index (BMI) of 22.2 and S.D. of 0.8 (48 Arab Muslims and 24 Jewish). They were recruited in a snowball method. Path analysis and principal component analysis are used in order to identify the interrelations among the physiological, cognitive and emotional indices and the directions of these interrelations. According to the findings, the Autonomic Nervous System (ANS), as measured by Heart Rate Variability (HRV) and primarily the parasympathetic tone (High frequency-HF) is the pivotal mechanism that modulates emotional and cognitive responses to environmental nuisances. The ANS response precedes and may trigger the emotional and the cognitive responses, which are only partially interrelated. It appears that the autonomic balance measured by Standard Deviation of NN interval (SDNN) and HF, the cognitive index of restoration and the emotional indices of discomfort and relaxation are closely interrelated. These seemingly disparate operands work together to form a comprehensive underlying network that apparently causes stress and risk to health in urban environments while restoring health in green environments.

## 1. Introduction

The notion that urban environments are highly stressful and that green environments have restorative power is widely accepted in the literature [1,2,3,4,5,6,7]. Urban environments stimulate chronic stress associated with insufficient recovery [8]. It is argued that stress is endemic in cities, regardless of exposure to any specific stressor. Staats (2012) relates this argument to a long tradition of research that emphasizes the stressful environments of cities [9]. He demonstrates a consistent line of argument evolving from Wirth’s seminal work on urbanism as a way of life [10], through the Chicago school argument of anomie [11], Simmel’s argument of strangerness in capitalist cities [12], and Milgram’s [13] and Lofland’s [14] argument of over stimulation in cities. According to these studies, large number of strangers concentrated in crowded, overstimulating environments characterizes modern cities exposing their residences to stressful situations.

Contemporary studies prove that long exposure to urban stresses is associated with cardiovascular diseases, obesity, immunological, gastroenterological, neurological and mental disorders [15,16,17]. Few studies also relate to social loads as sources of stress and risk to health [18]. Overload, fear of violence and discrimination are mentioned as major social stressors [19,20,21].

In studies that investigate the stresses associated with urban environments and the recovery generated by green areas, three groups of stress indicators are used. One set of studies focuses on the physiological effects of urban nuisances (such as heat load, air pollution, noise and discrimination) on the human ANS [22,23], with ANS disruption leading to and associated with stress and risk to health [24,25,26]. Many of these studies make use of time and frequency domain indices of heart rate variability (HRV) [27,28,29,30,31].

A second set of indices examines the cognitive effects of urban and green environments on human stress and health risk. The most commonly used indices are the perceived restoration potential index and memory tests [32,33,34,35]. A third set of indices focuses on the use of emotional indices such as positive and negative emotions index, sense of discomfort, cheerfulness and relaxation, among others [36,37,38,39,40]. The majority of studies use a single human response measure from one of these three domains. As a result, it is difficult to compare the findings of different studies, to elucidate the full range of risks posed to humans in urban environments, and to identify effective means of increasing the restorative power of urban and green environments. This type of single-measure methodology also falls short of ensuring convergent validity.

The use of stress indices is determined by three different theoretical approaches to understand the effects of exposure to environmental factors in cities and green environments. First, the psychophysiological theory argues that exposure to restorative environments is associated with improved ANS balance and an associated decreased psychological stress [41]. According to this theory, the physiological and the emotional aspects of human responses to environmental nuisances are intimately linked.

Second, the perceived restoration theory argues that some environments require directed attention, resulting in fatigue and a reduction in cognitive capabilities, whereas others, such as green environments, evoke indirect attention, resulting in restoration [42,43,44,45]. Restoration by undirected attention, according to this theory, results from fascination, a sense of being away, coherence, and compatibility [46].

Advocates of both theories criticize the opposing theory for developing a one-sided framework that focuses on either the cognitive or emotional aspects, ignoring the complex relationships between emotional, cognitive, and behavioral response mechanisms considered in psychology [44]. Scholars such as Hartig (2004) and Von Lindern et al. (2017) have proposed that the physiological and emotional aspects are inextricably linked [47,48]. Other researchers discovered a link between cognitive functioning and ANS balance as measured by HRV [49,50]. HRV was regarded as a passive indicator of emotional and/or cognitive functioning in both cases. Furthermore, no complete isomorphism between these three aspects of underlying restoration was expected, and their relationships remained ambiguous [9,51].

Attempts to improve our understanding of the interrelations among the responses of the three aspects to environmental nuisances have been reported. Learning from other contexts, it become clear that the three aspects are interconnected in their basic definitions. Scherer (1984) defines emotions as being constituted of several components including a cognitive and a physiological one [52]. Despite this, recent studies reject the classical assumption of a perfect harmonious coordination among the three aspects. Mauss et al. (2005) show that they are only partially correlated [53]. From Scherer’s (2010) study on emotions, it may be deduced that emotions stimulate cognitive processes that once they stabilize, they also affect physiological processes. Mendes (2016) adds that associations between emotions and physiological responses are mediated by various cognitive and bodily responses [54]. From this logic it may be deduced that the emotional responses appear first leading to cognitive processing of these feelings and resulting in physiological symptoms.

One of the few studies on the environmental effects on stress and risk to health focused on the effects of leaving the house to selected urban environments on cognitive, emotional and physiological responses, identifying several relevant indices that tapped into the three suggested mechanisms [55]. Reaching out to outdoor environments, particularly parks, was found to affect participants’ mood, HRV, and cognitive responses regardless of ethnic affiliation. However, because this study did not examine the interrelationships between the indices, it cannot support the validity of the psycho-physiological and perceived restoration theories.

Former theories of biophilia and stress recovery referred to these three domains but only with the introduction of the neurovisceral integration model over the last decade support the argument that emotion, cognition, and the ANS are intricately linked in response to environmental exposure [56]. It also suggests that HRV is more than just an indicator of emotional and cognitive states, but rather an active primary autonomic response that affects and possibly modulates emotional and cognitive states [57,58]. Fiol-Veny et al. (2019) [59], on the other hand, demonstrate that emotions influence HRV, implying a bidirectional relationship between emotional regulation and the cardiac system.

The goal of the present study is to analyze the inter-relations among nine of the more frequently used indices of response to environmental challenges that represent the emotional, cognitive and physiological reaction to environmental challenges. Such conceptual mapping may support one of the three proposed theories: the physiological-emotional, the cognitive and the neuro-visceral integration model. It may highlight how cognitive and emotional aspects reinforce each other and to what extent the emotional and the cognitive aspects are affected by or effect the ANS physiological system. The study may point to indices that best represent human responses to environmental challenges in cities. The study may also contribute to a better understanding of the complexity of the effects of human environmental experience on human well-being. In this study, our focus is on the coping mechanism as reflected by the emotional, cognitive and physiological aspects and not on the effects of different environments on stress and risk to health. Therefore, we do not distinguish between ethnicities and environments but we analyze the whole data set without accounting for ethnicity as we did in former studies [21,55].

In the present pilot study, we ask the following questions:What are the inter-correlations among indices of the emotional, cognitive and physiological aspects of human coping with exposure to urban daily environments?What are the dominant directions of these interrelation?

To what extent are the psycho-physiological, the attention restoration theory or the neuro-visceral integration theory empirically supported by the relationships between these respective domains

## 2. Methods

In the method section we present the study population, the procedure of running the field test and the methods of analysis.

### 2.1. Research Participants

The study is based on a relatively small sample. Therefore, we consider it as a pilot study. Data were collected from 72, middle class, healthy, young, non-smoking, and non-medicated married women aged 20 to 35 who took part in an ecological study [21]. They were recruited using a snowball procedure. out of them 43 percent having high school education and the rest post graduate education Seventy percent of them declared they had a fair economic status. About 60 percent of them had 1–2 children and their average BMI was 22.2 (0.8 S.D.) Almost all of them worked and took major responsibility of the children in their families. Women were tested since they are considered more sensitive to environmental conditions [60] and their role as mothers makes them of particular interest in the investigation of the relationship between environment and wellbeing. Since for the purpose of this study the women affiliation was not relevant, they were grouped into one cohort for the purpose of the present study. The larger sample size therefor adds to the statistical power of the present study. All women lived in the test region. The project was approved by the Ethics Committee of the University of Tel Aviv, Israel. Each participant signed a consent form that included a thorough explanation of the research objectives and procedure. For each group, questionnaires were administered in Arabic and Hebrew.

The study area is in the lower Galilee, in northern Israel, and is centered on two neighboring towns, Afula and Nazareth. According to Koepen, both cities have a Mediterranean climate [61]. The cities are surrounded by mountains in such a way that most places in these cities have open views to the horizon. Nazareth, however, is 200 m higher in elevation. There are town parks and several smaller parks in both cities, with Afula having more greenery than Nazareth. We chose three types of outdoor environments that together represent the range of typical urban environments from the more crowded to quiet residential areas and to urban parks. The city center was highly crowded with transportation, markets and shops and 4–5 stores buildings. The residential areas were of 3–5 stores houses with green yards and few cars and pedestrian in the streets. The parks were city parks with trees that cover about half of the view to the city, grasses and children playground.

The underlying hypothesis of the original project was that each individual and possibly of a specific ethnic affiliation experiences differently urban environments with respective physiological and mental response. However, since our main interest in the present study was to elucidate the interrelations among different reactivity domains (emotional, cognitive and physiological) in human beings, for the present analysis both ethnic groups and environments were collapsed into one cohort and undifferentiated environment to increase the statistical power.

### 2.2. Procedure

The fieldwork was divided into twelve sessions with 6 participants in each session (6 participants × 12 sessions). Each participant visited all four environments (home and a park, residential site and city center in their hometown). All of the subjects in all of the sessions followed exactly the same route and siting sites. One of the researchers visited each participating woman at home one day before the experiment to explain the goal of the study to sign an agreement form, and to fill out a questionnaire on sociodemographic characteristics, supply the devices, and explain how to use them. The experiment began at home and progressed to the three outdoor environments. They visited the outdoor environments on the same day in a random order, with a 15 min break in an air-conditioned car set to 22 °C. The researcher followed the subjects along the route inspecting the devices and ascertaining adherence to the protocol.

The sessions were held between June 2015 and February 2016. The devices were calibrated prior to the sessions per the producer’s instructions [29].

The cognitive and emotional tests were given to each participant near the end of their 35 min stay at home and in each outdoor environment while they were sitting on the grass or on a bench. The outdoor experiments started at 12:00 p.m. Women were asked to avoid interacting with other participating women in all environments. All of themeasurements were taken at the participant’s home and in each environment.

### 2.3. The Indices of Response to Environmental Challenges

We employed a well-documented set of indices to assess the emotional, cognitive, and physiological responses to urban exposure (Figure 1). We related these to four emotional indices: mood, discomfort, cheerfulness, and relaxation. Two cognitive indices were used: The Self Perceived Restoration Scale (PRS) and backward number memory. We chose the main time and frequency domain indices of HRV as physiological indices.

#### 2.3.1. Physiological Indices

Several physiological measurements are used in studies on stress and risk to health in urban environments. Among them skin conduction response, salivary cortisol, brain activity, gait pattern and Heart Rate Variability. HRV indices were chosen for this study since they are a well-documented method, accessible for use in trackable outdoor environments studies and cheap to implement. In addition, the results are in correlation with result extracted from the other methods. Several studies showed that HRV supply only a crude estimation of stress and risk to health suggesting supplementing HRV studies with one of the other mentioned indices. Bernston et al. (1991) [62] show, for example, that the sympathetic and the parasympathetic tones do not lie along a single continuum but along a two-dimensional space due to the either co-active or reciprocal coordination between the sympathetic and the parasympathetic operation. However, recent meta-analyses confirm that HRV adequately reflects the ANS balance and activity in environmental studies [63,64]. HRV was monitored and stored by Polar 810i trackable device that recorded heart bits for 35 min in each of the studied environments: home as an indoor measurement; park; residential and city center as outdoor environments. They sat down for the 35 min, observing the view from the bench and they answered a short questionnaire that included five questions about their experience in the environment.

From these recordings, a total of seven five-minute segments were sequentially analyzed. Both time and frequency domain were included [65]. Of the frequency domain indices, High Frequency power (HF) which ranges between 0.15 and 0.4 Hz, is considered as a marker of the parasympathetic tone, which is primarily affected by respiration. It correlates with the respiratory sinus arrhythmia.

Low Frequency power (LF) which ranges between 0.04 and 0.15 Hz, is activated by both the parasympathetic and the sympathetic systems. LF may increase as a result of physical or psychological stress. These indices were empirically validated [64,66]. The regulation of HRV is immediate. Sympathetic input leads to changes in heart rate, with a peak after about 4 s and a possible recovery latency of 20 s following the stimulus. The parasympathetic system responds within 0.5 s and can return to baseline within 1 s after the stimulus ends. LF/HF ratio represents the ANS balance, and it reflects the resulting general ANS activity at the time of measurement.

Time domain indices measure variation of the intervals between consecutive cardiac cycles, which is the standard deviation of consecutive heart bit picks of five minutes intervals. SDNN, the standard deviation of the average NN intervals, is the most general and frequently used time domain HRV index. For signal analysis, including artifact removal, the Kubios HRV software version 2 (www.kubios.uku accessed on 29 March 2022) was used.

#### 2.3.2. Emotional Measurements

The participants filled out a questionnaire that assessed their mood, cheerfulness, sense of relaxation, and general discomfort in relation to each of the environments they visited. Mood was assessed employing the Positive and Negative Affect Schedule (PNAS) [67,68]. Participants rated their state of mind using a list of 10 positive and 10 negative moods in respect to each of the environments they visited. They rated their moods on a 5-point Likert scale, ranging from 1 (very slightly) to 5 (extremely). Alpha Cronbach reliability test reached 0.94. Relaxation and cheerfulness were evaluated based on a modified semantic differential method that was found to be accurate by Osgood (1957) [69]. The participants evaluated their state of cheerfulness and relaxation, while staying in each of the tested environments. The items of the questionnaires were rated on a 7-point Likert scale from 0 (not at all) to 6 (extremely). Sense of discomfort was assessed using a color analog scale (CAS) that measured a general sense of discomfort in the visited environment (e.g., the participants were asked the question “To what extent do you feel general discomfort at this moment?”). The scale ranged from 0 to 100, and from dark green (Sense of comfort) to dark red (Sense of discomfort)) [70,71]. To ascertain the content validity of the employed instruments, the binary correlations among the emotional indices were calculated. The correlations ranged between 0.5 and 0.8, and thus were considered suitable for further use in the analysis.

#### 2.3.3. Cognitive Measurements

Self-perceptions of a restorative experience were evaluated based on the ‘perceived restrictiveness scale’ (PRS). The scale is composed of 26 statements concerning participants’ experience of environments’ restorativeness. Each statement was rated on a 7-point Likert scale of agreement ranging from 0 (highly disagree) to 6 (highly agree) [72]. The statements related to the four components of the scale: ‘Being away’, ‘Fascination’, ‘Coherence’ and ‘compatibility’. An Alpha Cronbach reliability test was calculated for the total sum and for each of the components of PRS, reaching values above 0.82. Working memory was evaluated based on backwards digit-span task (BDSP). Participants were asked to memorize and repeat in reverse of three to nine digits long sequences. After two consecutive errors, the test was terminated. Regardless of the number of digits in the recalled sequences, the participants’ scores equaled the number of correctly recalled sequences [73].

## 3. Analysis

The analysis started by standardizing the nine physiological, emotional, and cognitive indices [(I − I_mean_)/I_mean_]. At the second stage, a path analysis was applied. Structural Equation Modeling (SEM) using an AMOS SPSS path model was run to calculate the coefficients among the indices. The model was run several times, changing the direction of the paths between each couple of indices and searching for the dominant directions of paths till the model with the best t was obtained. In the final model, the dominant paths between each two indices were chosen. Lastly, a component principal analysis was applied in order to identify clusters of variables that are interrelated. The two methods complement each other. While AMOS path analysis allows us to calculate one directional paths among single indices, component analysis allows for clustering several indices into higher level components as a basis for the verification of the aforementioned three competing theoretical frameworks.

## 4. Results

The nine indices differ in their distribution. Some have a wide distribution (relaxation and cheerfulness), indicating high sensitivity to environmental challenges, whereas others are less sensitive to the individual responses to environmental challenges (discomfort). With the exception of LnLF, the differences in the standardized indices of high and low levels of response to environmental challenges are significant in all measurements (Table 1). Following the path analysis, it was evident that both LF and LF/HF were redundant.

The paths among the three groups of indices were calculated in both directions between each couple of indices. In Figure 2, the more dominant directions are presented. The first conclusion from the analysis is that SDNN and HF dominate the model because they initiate all of the paths that lead to the cognitive and emotional indices. This means that the autonomic indices, particularly the parasympathetic branch, will be the first to respond with the shortest latency, possibly eliciting a response from the other emotional and cognitive networks. Therefore, the AMOS SPSS path model was built around the physiological indices of SDNN and HF (Figure 2). The path between SDNN and HF reached a level of 0.95, which means that they measure similar aspects of human response to environmental challenges. Despite this, they differ in how they are linked to emotional and cognitive indices. SDNN dominates the cognitive indices with paths values of 0.83 and 0.73, respectively, while HF partly affects the cognitive index of backward memory. At the same time, SDNN and HF affect the emotional indices, although with a smaller magnitude compared to the cognitive indices. SDNN primarily affects discomfort and has no relationship to relaxation. HF primarily affects relaxation and positive emotions, with a lesser impact on cheerfulness and discomfort. Despite the fact that SDNN and HF have extremely strong coefficients, they are associated with cognitive and emotional responses in different ways. While SDNN has a strong positive effect on cognitive indices, both SDNN and HF have a partial effect on emotional indices. Furthermore, while a decrease in SDNN appears to enhance discomfort, an increase in HF is followed by more relaxation and positive emotions.

In testing associations between the emotional and the cognitive indices, as shown in Figure 3, while strong paths are evident between restoration and all of the emotional indices, no direct association is recorded between backward memory and the emotional indices other than the positive effect of relaxation and the negative effect of discomfort on backward memory. In other words, while discomfort has a significant negative effect on restoration, in turn, restoration positively affects the rest of the emotional indices.

In clustering the indices, two components appear to explain 83% of the variability in coping with environmental challenges (Table 2). Component one includes the physiological and the cognitive indices (Back; restoration; HF; SDNN). This factor explains 47% of the variability in the coping mechanisms employed in the face of environmental challenges. Component 2 includes the emotional indices (Positive emotions; Cheerfulness; Relaxation), but discomfort is more associated with component 1’s physiological indices (Figure 4). Component 2 accounts for 36% of the variation in coping with environmental challenges.

## 5. Discussion

The three research questions are all intertwined. Regarding the first question, it appears that the physiological, emotional, and cognitive aspects of dealing with environmental challenges in cities are all intertwined. With regards to to the second question, all of the paths between the physiological and the emotional or the cognitive indices are dominated by one direction, from the physiological to the cognitive or the emotional indices. According to the findings, higher levels of ANS activity, particularly the parasympathetic branch (HF), help to improve emotional regulation and cognitive functioning. In more detail, HF and SDNN have an impact on all of the emotional indices. At the same time, SDNN affect the two cognitive indices while HF affects only backward memory. This result contradicts the conclusions of Scherer (2010) [52] and Mendes et al. (2016) [54] concerning the primacy of the emotional aspect in initiating responses to environmental challenges. Instead, it supports the neurovisceral integration model [58]. It seems that more research that focuses on the forms of integration between the emotional, cognitive and emotional aspects.

This is consistent with a plethora of studies on the effects of urban environments on human well-being, but it adds new insight to the interrelations among the emotional, cognitive and physiological responses. The majority of these studies concentrated on the effects of urban and green environments on one of the operative systems: physiological, emotional, or cognitive [9,43,45]. This is also consistent with the findings of Saadi et al. (2020) [55], who demonstrated the validity of all three aspects of emotional, cognitive, and autonomic responses to environmental challenges without investigating their interrelations and directions of effects. It does not, for example, emphasize the ANS’s status as an active indicator of emotional and cognitive states or as an independent factor modifying the processing and output of the emotional and cognitive systems. The contribution of this study is in highlighting the interrelations among the indices. Furthermore, the analysis exposes the dominance of the physiological indices over the cognitive and the emotional ones.

The autonomic indices themselves are only partially associated. Due to low internal variability, LF is not associated with any of the other indices. HF mainly affects relaxation and positive emotions, as well as all the other indices, except for restoration. The effect of SDNN on restoration and backward memory is dominant. SDNN withdrawal affects discomfort and, to a lesser extent, cheerfulness and positive emotions. The predominant fluctuations affecting ANS balance are attributed to the more sensitive parasympathetic response (either enhancement or withdrawal) in the face of both emotional and cognitive challenges. This finding has been discussed extensively in a previous review [74].

With regards to the third question, our study does not contradict the psycho-physiological and the attention restoration theories. Both are linked to the physiological, i.e., autonomic indices. However, our findings suggest that the disclosive discrete approach, which relies on one of these theories to explain responses to environmental challenges, may be oversimplified. The findings of this study point to an interactive multidimensional mechanism, with the autonomic system playing a dominant role in modulating the other subserving mechanisms, as previously proposed [56,57,58].

The ANS, specifically as reflected by HRV, has previously been shown to play a dominant role in emotional and cognitive regulation, as well as positive mood regulation [56]. Furthermore, it has been reported that self awareness, a cognitive process and the activity of the brain centers sub-serving cognitive processes are highly related to fluctuations in HF HRV and, in fact, likely precede the subjects’ response to cognitive challenges [75].

In our case, the paths between the physiological and the cognitive indices are stronger than the paths between the physiological and the emotional indices. The physiological and the cognitive indices are included in one component in the principal component analysis, and they take a stronger share in the explanation of variability in responses to environments (47%) compared to the share of the emotional component (36%), while the emotional indices are combined into a separate component. This ending emphasizes the critical role of cognitive processes in dealing with environmental challenges, as several scholars have argued [46,47,48]. At the same time our results point to a more moderate association between physiological and emotional processes, compared to what was previously argued by scholars who followed Ulrich [41].

An additional comment is the fact that discomfort that was considered to represent an index of emotional response was found to belong to the cognitive component in the component analysis.

## 6. Study Limitations

The most obvious limitations of the study are two: First, the only physiological indices included in the study related to HRV. Other complementary methods such as skin conduction, salivary cortisol and brain tests were not included. This is due to lack of availability and the inflexibility of the use of some methods in outdoor tracking methods used in our study. Second, our sample includes young, middle class, healthy nonmedicated women. Wider studies employing a larger sample size that will be more representative of the population understudy are needed.

## 7. Conclusions

The present study exposes an intricate relationship between the responses of human beings to their environment, particularly to a restorative experience. Emotional, cognitive and autonomic (physiological) underlying mechanisms are activated and affect each other. Furthermore, the theoretical equipotential bidirectional relationship between HRV and emotional regulation [76] is not confirmed by our study. In fact, the results of the path analysis, which used either the cognitive and emotional or the HRV indices as independent variables, revealed that the latter had a more robust effect on the former outcomes. Hence, we present the best model only. From this model, one may conclude that the ANS response precedes processing activity related to tasks tapping emotional and cognitive faculties. The dominance of physiological indices over emotional and cognitive indices, as well as the complex relationships between the indices, appear to call for additional research on the operation of the three systems (the autonomic, the emotional and the cognitive) in coping with urban and green environments.

## Figures and Tables

**Figure 1 ijerph-19-08190-f001:**
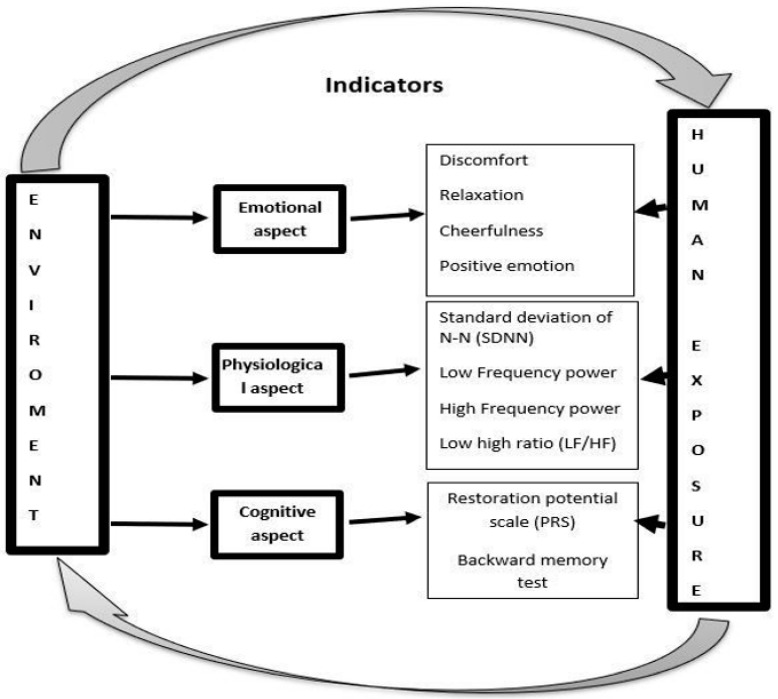
The indices of a general model of environmental stress.

**Figure 2 ijerph-19-08190-f002:**
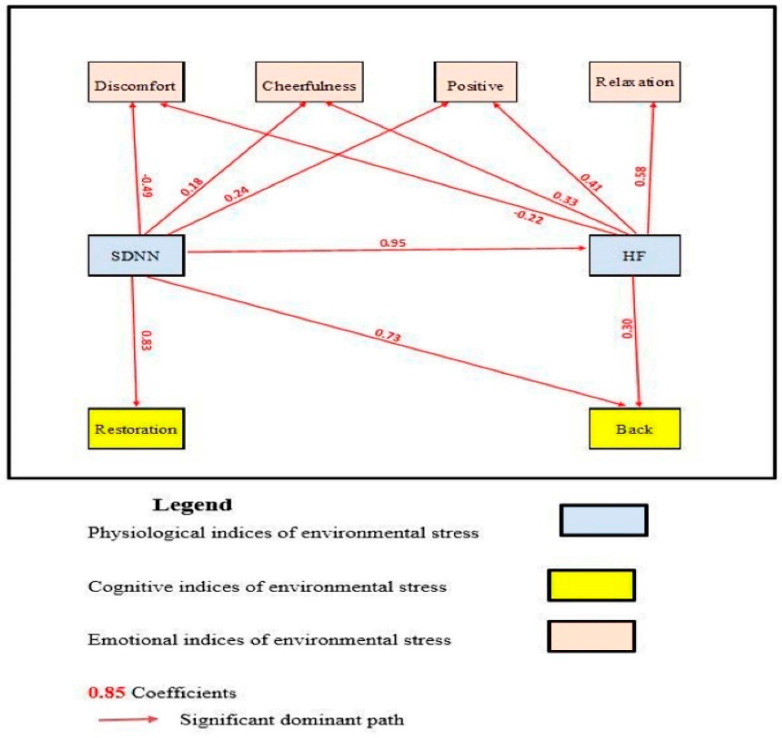
An AMOS SPSS path model of indices of coping with environments.

**Figure 3 ijerph-19-08190-f003:**
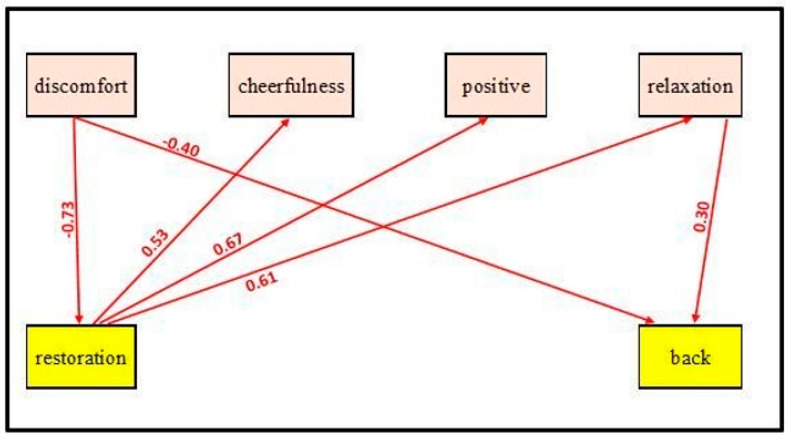
An AMOS SPSS path model of cognitive vs. emotional indices of coping with Environments.

**Figure 4 ijerph-19-08190-f004:**
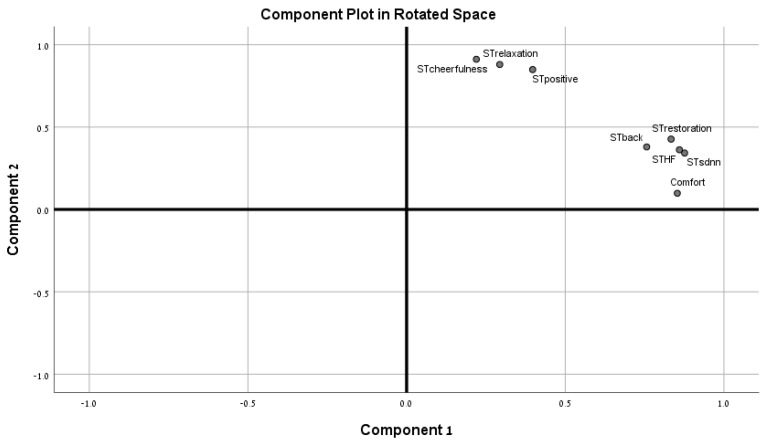
Components plot.

**Table 1 ijerph-19-08190-t001:** Mean levels of the standardized values of Low and high Coping indices.

Standardized Indices	Group				*p*-Value	Difference
	Low		High			
	Mean	SD	Mean	SD		
Discomfort	−0.11	0.40	−0.58	0.45	<0.001	0.47
Relaxation	−0.80	0.25	0.95	0.32	<0.001	−1.75
Cheerfulness	−0.72	0.29	0.80	0.28	<0.001	−1.52
Positive	−0.39	0.23	0.55	0.16	<0.001	−0.94
Restoration	−0.23	0.12	0.70	0.30	<0.001	0.93
Back	−0.97	0.09	−0.72	0.07	<0.001	0.25
LnSDNN	−0.22	0.15	0.53	0.25	<0.001	−0.75
LnLF/HF	0.13	0.31	−0.63	0.35	<0.001	−0.76
LnLF	−0.05	0.17	−0.07	0.28	0.547	−0.02
LnHF	−0.45	0.22	1.08	0.65	<0.001	1.53

Low represents the average value for the 25 percent with lowest values and high represents average values for the 25 percent with highest values.

**Table 2 ijerph-19-08190-t002:** Rotated Component Matrix.

	Component	
	1	2
STrelaxation	0.293	0.880
STcheerfulness	0.293	0.912
STpositive	0.397	0.849
STsdnn	0.876	0.343
STHF	0.860	0.362
STrestoration	0.833	0.427
STback	0.757	0.380
Comfort	0.853	0.098

Extraction Method: Principal Component Analysis; Rotation Method: Varimax with Kaiser Normalization; Rotation converged in 3 iterations.

## Data Availability

Data can be supplied by the corresponding author in response to request.

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
