# Peer review of "Women Emotional, Cognitive and Physiological Modes of Coping with Daily Urban Environments: A Pilot Study"

_ijerph, 2022, doi:10.3390/ijerph19138190_

Round 1

Reviewer 1 Report

Thank you for the opportunity to review this interesting and well-articulated study focussing on a sample of Middle Eastern women’s responses to different urban environments. The study itself is timely, and it’s great to see attention being given to potentially under-studied locations and demographics within this field.

The introduction is good, and provides good justification for the study. While the methods appear to be sound, further information is required for the reader to fully evaluate and understand the processes and results. My main suggestions for improvement are:

  • Ensure your research questions match the current study. The current questions are really ambitious, which is great, but the current study focusses on a small sample in Israel- can this (understudied and very necessary sample) address these specific questions?
  • There is also no mention of the chosen demographic in the Abstract or Introduction. As this is such a key component of the study, I am surprised this is not discussed at all.
  • Start your methods by outlining the research process. I found the current structure quite confusing, as the measures used are described in detail within the introduction, so it’s not entirely clear whether this relates to your study or is a description of secondary research. Please move this section to the Methods. I’d suggest you start by providing an overview of the methods before moving onto the specifics of the data collection.
  • On the same theme, please provide full details of the sampling and data collection methods. Please explain why you selected this demographic, this sample size, how participants were recruited, whether participants were aware of the study aims, and how the data was collected. Please also describe the format of the sessions, when they were run. Further information is needed on the different environments, why these were selected, whether participants in different locations visited different environments, and how these were controlled. Photos of the different locations would be incredibly helpful.
  • Was any control data on participants collected? If not, why not? This should also be included in the limitations.
  • Is data included before and after the different environments? If so, this should be included.
  • In the results, please begin by providing full descriptive statistics of the study and participants

Further comments:

  • Please ensure you define all acronyms the first time they appear
  • Please provide citations of all software and explanations of all methods
  • Lines 36-38: these need to be properly explained if you want to include them
  • Line 45: the discussion of methods is good, but what have these studies found?
  • Line 47: could you provide examples of these nuisances?
  • Line 61: what about other major theories, namely biophilia and the Stress Recovery Theory? The explanation reads as though the theories you discuss are the only ones.
  • Line 151: should this be ‘sat down’?
  • Line 208: what is the context of ‘Muslim women being more conservative’?
  • Please thoroughly proof read the manuscript to check for typos

Author Response

Responses to reviewer 1

Dear reviewer. As you can see, we made some major revisions in the article according to your valuable suggestions and comments. We believe the main message of the article is now clearer and the methodological aspects more accurate.  In order to facilitate easy reviewing the enclosed list of responses ordered as for the comments, is followed by the text appearing in the present revision with the exact lines in the text also provided. 

  1. Adjust the research questions to the current research:

Many of the comment are assuming that we seek to compare between ethnic groups, which we are not. We understand the title of the article is confusing. Therefore, we revised the title and removed the reference to   the ethnic groups. Such comparisons are the focus of several of our former articles. In this article, we focus on the human mechanism of cognitive, emotional and physiological coping with environments. We have no reason to assume that these mechanisms are effected by ethnicity. Therefore, we collated the sample into one cohort benefiting from the larger sample a greater statistical power .  ( pp.3,  lines 128-132)  

"In this study our focus is on the   coping mechanism as reflected by the emotional, cognitive and physiological aspects and not on the effects of different environments on stress and risk to health. Therefore, we do not distinct between ethnicities and environments but we analyze the whole data set with nort accounting for ethnicity as we did in former studies [21, 55] "  

In order to acknowledge the incipient nature of this study using a  small sample size  the wording  "a pilot study" is now included in the title  .We further relate to this point at the end of the introduction.

The new title is: "Women emotional, cognitive and physiological modes of coping with daily urban environments: A pilot study".

Extending the sample characteristics in the abstract and the introduction:

We added to the abstract information on the sample in lines 19-23.

"The data were collected in the following four environments: Home, park, city center and residential area. The participants were 72 healthy, middle class mothers with either high school or postgraduate education. They were in their fertile age (20-35) with average Body Mass Index( BMI ) of 22.2 and S.D. of 0.8 (48. Arab Muslims and 24 Jewish). They were recruited in a snowball method"

We also added information on the socio demographic characteristics of our sample to the methodological section (Lines 146-150).

" They were recruited using a snowball procedure. Out of them 43 percent having high school education and the rest post graduate education.  Seventy percent of them declared they had a fair economic status. About 60 percent of them had 1-2 children and their average BMI was 22.2 (0.8 S.D) . . Almost all of them worked and took major responsibility on the children in their families. Women were tested since they are considered more sensitive to environmental conditions [72]  and their role as mothers makes them of particular interest in the investigation of the relationship between environment and wellbeing. Since for the purpose of this study the women affiliation was not relevant they were grouped into one cohort for the purpose of the present study. The larger sample size therefor adds to the statistical power of the present study"

In addition we extended the information on the environments under study (lines 165-170)

"We chose three types of outdoor environments that together represent the range of typical urban environments from the more crowded to quiet residential areas and to urban parks. The city center was highly crowded with transportation, markets and shops and 4-5 stores buildings. The residential areas were of 3-5 stores houses with green yards and few cars and pedestrian in the streets. The parks were city parks with trees that cover about half of the view to the city, grasses and children play ground. .

The underlying hypothesis of the original project was that each individual and possibly of a specific ethnic affiliation experiences differently urban environments with respective physiological and mental response. However, since our main interest in the present study was to elucidate the interrelations among different reactivity domains ( Emotional , cognitive and physiological ) in human beings, for the present analysis both ethnic groups and environments were collapsed into one cohort and undifferentiated environment to increase the statistical power. "

  1. Indices should be part of the methodological section.

We transferred the extended  section on the indices to the methodology.

" The indices of response to environmental challenges

We employ a well-documented set of indices to assess the emotional, cognitive, and physiological responses to urban exposure (Figure 1). We relate to four emotional indices: mood, discomfort, cheerfulness, and relaxation. Two cognitive indices were used: The Self Perceived Restoration Scale (PRS) and backward number memory. We chose the main time and frequency domain indices of HRV as physiological indices.

Figure 1. The indices of a general model of environmental stress.

2.3.1. Physiological indices

Several physiological measurements are used in studies on stress and risk to health in urban environments. Among them skin conduction response, salivary cortisol, brain activity, gait pattern and Heart Rate Variability. HRV indices were chosen for this study since it is a well documented method, accessible for use in trackable outdoor environments studies and cheap to implement. In addition, results are in correlation with result extracted from the other methods. Several studies showed that HRV supply only crude estimation of stress and risk to health suggesting supplementing HRV studies with one of the other mentioned  indices. Bernston et al.  (1991)[60] show for example, that the sympathetic and the parasympathetic tones do not lie along a single continuum but along a two dimensional space due to the either co-active or reciprocal coordination between the sympathetic and the parasympathetic operation. However, recent meta-analyses confirm that HRV adequately reflects the ANS balance and activity in environmental studies[61], [62]. HRV was monitored and stored by Polar 810i trackable device that recorded heart bits for 35 minutes in each of the studied environments: home as an indoor measurement; park; residential and city center as outdoor environments. They set down for the 35 minutes observing the view from the bench and they answered a short questionnaire that included five questions about their experience in the environment.

From these recordings seven, five-minute segments were sequentially analyzed. Both time and frequency domain were included[63]. Of the frequency domain indices, High Frequency power (HF) which ranges between 0.15 and 0.4 Hz, is considered as a marker of the parasympathetic tone, which is primarily affected by respiration. It correlates with the respiratory sinus arrhythmia. Low Frequency power

(LF) which ranges between 0.04 and 0.15 Hz, is activated by both the parasympathetic and the sympathetic systems. LF may increase as a result of physical or psychological stress. These indices were empirically validated[62], [64].The regulation of HRV is immediate. Sympathetic input leads to changes in heart rate, with a peak after about 4 s and a possible recovery latency of 20 s following the stimulus. The parasympathetic system responds within 0.5 s and can return to baseline within 1 s after the stimulus ends. LF/HF ratio represents the ANS balance and it reflects the resulting general ANS activity at the time of measurement.

Time domain indices measure variation of the intervals between consecutive cardiac cycles, which is the standard deviations of consecutive heart bit picks of five minutes intervals. SDNN, the standard deviation of the average NN intervals, is the most general and frequently used time domain HRV index. For signal analysis, including artifact removal, the Kubios HRV software version 2 (www.kubios.uku. ) was used.

2.3.2. Emotional measurements

The participants filled out a questionnaire that assessed their mood, cheerfulness, sense of relaxation, and general discomfort in relation to each of the environments they visited. Mood was assessed employing the Positive and Negative Affect Schedule (PNAS)[65], [66]. Participants rated their state of mind using a list of 10 positive and 10 negative moods in respect to each of the environments they visited. They rated their moods on a 5-point Likert scale, ranging from 1 (very slightly) to 5 (extremely). Alpha Cronbach reliability test reached 0.94. Relaxation and cheerfulness were evaluated based on a modified semantic differential method that was found to be accurate by Osgood (1957)[67]. The participants evaluated their state of cheerfulness and relaxation, while staying in each of the tested environments. The items of the questionnaires were rated on a 7-point Likert scale from 0 (not at all) to 6 (extremely). Sense of discomfort was assessed using a color analog scale (CAS) that measured a general sense of discomfort in the visited environment (e.g. the participants were asked the question “To what extent do you feel general discomfort at this moment?”). The scale ranged from 0 to 100, and from dark green (Sense of comfort) to dark red (Sense of discomfort) ([68], [69]. To ascertain the content validity of the employed instruments, the binary correlations among the emotional indices were calculated. The correlations ranged between 0.5 and 0.8, and thus were considered suitable for further use in the analysis.

2.3.3. Cognitive measurements

Self-perceptions of a restorative experience was evaluated based on the 'perceived restrictiveness scale' (PRS). The scale is composed of 26 statements concerning participants' experience of environments' restorativeness. Each statements was rated on a 7-point Likert scale of agreement ranging from 0 (highly disagree) to 6 (highly agree)[70]. The statements related to the four components of the scale: 'Being away', 'Fascination', 'Coherence' and 'compatibility'. An Alpha Cronbach reliability test was calculated for the total sum and for each of the components of PRS, reaching values above 0.82.Working memory was evaluated based on backwards digit-span task (BDSP). Participants were asked to memorize and repeat in reverse of three to nine digits long sequences. After two consecutive errors, the test was terminated. Regardless of the number of digits in the recalled sequences, the participants' scores equaled the number of correctly recalled sequences [71]."

  1. Expanding the information on the sampling procedure

Lines 146-147 we now acknowledge the small sample size. We mention that they were recruited with a snow ball procedure  and hence an homogeneous not representative of the population at large.

"They were recruited using a snowball procedure. Out of them 43 percent having high school education and the rest post graduate education . . Seventy percent of them declared they had a fair economic status. About 60 percent of them had 1-2 children and their average BMI was 22.2 (0.8 S.D) "

  1. Justify the sample

See lines 151-156 we justify the choice of young healthy women who are more sensitive compared to men to environmental conditions. Therefore, they fit better  a pilot study.

"Women were tested since they are considered more sensitive to environmental conditions [72]  and their role as mothers makes them of particular interest in the investigation of the relationship between environment and wellbeing. Since for the purpose of this study the women affiliation was not relevant they were grouped into one cohort for the purpose of the present study. The largerer sample size therefor adds to the statistical power of the present study."

  1. A set of methodological comments:

The participants as reported  were informed about the goals of the study (1830-186).

" One of the researchers visited each participating  woman at home one day before the experiment to explain the goal of the study  to sign an agreement form… ,"

We added some details on the characteristics of the environments (lines 165-171).

" We chose three types of outdoor environments that together represent the range of typical urban environments from the more crowded to quiet residential areas and to urban parks. The city center was highly crowded with transportation, markets and shops and 4-5 stores buildings. The residential areas were of 3-5 stores houses with green yards and few cars and pedestrian in the streets. The parks were city parks with trees that cover about half of the view to the city, grasses and children play ground."

 We report now  that the subject visited and were sitting in the same sites while being tested (183-184).

"All subjects in all sessions followed exactly the same route and siting sites."

 The study dates are now provided (between June 2015 and Feb 2016 (192-193).

  1. Data before and after.

             Since this is not a before-after design no data related to any intervention or    

              point in time is provided.

  1. Descriptive data

We added socio-demographic information of the subjects at the beginning of the methodological section. This is in addition to table 1 that delineates the most relevant descriptive data of the studied indices. (lines 147-150)

"Out of them 43 percent having high school education and the rest post graduate education . Seventy percent of them declared they had a fair economic status. About 60 percent of them had 1-2 children and their average BMI was 22.2 (0.8 S.D) . . Almost all of them worked and took major responsibility on the children in their families. Women were tested since they are considered more sensitive to environmental conditions [72]"

  1. What does it means on lines 46-8?

We added a sentence on line 49-51

  1. Examples for nuisances:

Line 59 in the original text but now on line 56-57 we added examples.

(such as heat load, air pollution, noise and discrimination)

  1. What about biophilya and stress recovery theories?

We added a mention of biophilya and stress recovery theories on line111-118.

"Former theories of biophilia and stress recovery referred  to these three domains  but only with the  introduction of the neurovisceral integration model over the last decade  support  the argument that  emotion, cognition, and the ANS are intricately linked in response to environmental exposure [56]"

Reviewer 2 Report

This research seeked to examine interrelationships between physiological, cognitive, and emotional indices related to contextual environments. The manuscript is well written, starting with a through introduction of the various indices that have been developed and utilized in previous research, and continuing on with a clear methodology section, through analysis, and a conclusion. The strength in the manuscript lies in the fact that examining interrelationships of diverse physiological, cognitive, and emotional indices is novel in this area. Furthermore, the research was able to determine the strength and direction of the links between these indices, and from that link it to the predominant theories on exposure to green space (physiological -emotional theory and the cognitive theory of direct attention). As in my brief review, the authors could have added more general analyses on which environments provoked certain responses, and whether there are differences among Jewish and Arab women. It may be that the authors are planning to publish such findings in a separate manuscript, however the title gives the impression to the reader that this information will be presented in this manuscript. Besides these limitations, there were minor editorial errors which I have pointed to in my main review. Overall, this research is well thought and the manuscript is well written.

Minor editorial changes are needed, specifically:

p5, l224 - replace 'll'

p6, l260, replace 'rst'

p9, l353, replace 'nding'

p9, l362, there is mention to 2 limitations, but 'limitation' is singular - please rephase

p9, please clarify sentence starting on l377, 'From this model,....'

Author Response

Reviewer 2:

  1. Relate to ethnic differences

Ethnic differences are the focus of several of our former studies. In this study we focus on universal human coping mechanism with environments. This is beyond any differences among ethnicities or environments.

  1. We have corrected the minor typos.
